# Spatial and temporal clustering of typhoid fever in an urban slum of Dhaka City: Implications for targeted typhoid vaccination

**Faisal Ahmmed**[1]◔*, **Farhana Khanam**[1]◔, **Md Taufiqul Islam**[1], **Deok Ryun Kim**[2], **Sophie Kang**[2], **Md Golam Firoj**[1], **Asma Binte Aziz**[2], **Masuma Hoque**[1], **Xinxue Liu**[3], **Hyon Jin Jeon**[2,4], **Suman Kanungo**[5], **Fahima Chowdhury**[1], **Ashraful Islam Khan**[1], **Khalequ Zaman**[1], **Florian Marks**[2,4,6,7], **Jerome H. Kim**[2], **Firdausi Qadri**[1], **John D. Clemens**[1,2,8,9], **Birkneh Tilahun Tadesse**[2‡], **Justin Im**[10‡]

**1** International Centre for Diarrhoeal Disease Research, Bangladesh, Dhaka, Bangladesh, **2** International Vaccine Institute, Seoul, Republic of Korea, **3** Oxford Vaccine Group, Department of Pediatrics, University of Oxford, Oxford, United Kingdom, **4** Cambridge Institute of Therapeutic Immunology and Infectious Disease, University of Cambridge School of Clinical Medicine, Cambridge Biomedical Campus, Cambridge, United Kingdom, **5** ICMR- National Institute of Cholera and Enteric Diseases, Kolkata, West Bengal, India, **6** Madagascar Institute for Vaccine Research, University of Antananarivo, Antananarivo, Madagascar, **7** Heidelberg Institute of Global Health, University of Heidelberg, Heidelberg, Germany, **8** UCLA Fielding School of Public Health, Los Angeles, California, United States of America, **9** Vaccine Innovation Center, Korea University School of Medicine, Seoul, Republic of Korea, **10** RIGHT Foundation, Seoul, Republic of Korea

◔ These authors contributed equally to this work.
‡ BTT and JI also contributed equally to this work.
* faisal.ahmmed@icddrb.org

**Data Availability Statement:** De-identified analytical data set will be made available upon requests directed to the Institutional Review Board

## Abstract

### Background

*Salmonella enterica* serotype Typhi (*Salmonella* Typhi) causes severe and occasionally life-threatening disease, transmitted through contaminated food and water. Humans are the only reservoir, inadequate water, sanitation, and hygiene infrastructure increases risk of typhoid. High-quality data to assess spatial and temporal relationships in disease dynamics are scarce.

### Methods

We analyzed data from a prospective cohort conducted in an urban slum area of Dhaka City, Bangladesh. Passive surveillance at study centers identified typhoid cases by microbiological culture. Each incident case (index case) was matched to two randomly selected index controls, and we measured typhoid incidence in the population residing in a geographically defined region surrounding each case and control. Spatial clustering was evaluated by comparing the typhoid incidence in residents of geometric rings of increasing radii surrounding the index cases and controls over 28 days. Temporal clustering was evaluated by separately measuring incidence in the first and second 14-day periods following selection. Incidence rate ratios (IRRs) were calculated using Poisson regression models.

(IRB) coordinator, of the icddr,b, M.A Salam Khan, at salamk@icddrb.org, or at info@icddrb.org. Only after approval of a proposal data can be shared through a secure online platform. Approval of the proposal will be subject to scientific review by the institutional review board at icddr,b. Sharing of data will also be subject to the published data access rules of the icddr,b. The requestor will need to sign a standard data access agreement required by the icddr,b.

**Funding:** This work was supported by the Bill and Melinda Gates Foundation (INV-025388). The icddr,b is grateful to the Government of Bangladesh and Canada (Global Affairs Canada). The IVI is supported by the governments of Korea, Sweden, India, Finland, Denmark, the Philippines and Thailand. The funders had no role in study design, data collection and analysis, decision to publish, or preparation of the manuscript.

**Competing interests:** The authors have declared that no competing interests exist.

## Results

We evaluated 141 typhoid index cases. The overall typhoid incidence was 0.44 per 100,000 person-days (PDs) (95% CI: 0.40, 0.49). In the 28 days following selection, the highest typhoid incidence (1.2 per 100,000 PDs [95% CI: 0.8, 1.6]) was in the innermost cluster surrounding index cases. The IRR in this innermost cluster was 4.9 (95% CI: 2.4, 10.3) relative to the innermost control clusters. Neither typhoid incidence rates nor relative IRR between index case and control populations showed substantive differences in the first and second 14-day periods after selection.

## Conclusion

In the absence of routine immunization programs, geographic clustering of typhoid cases suggests a higher intensity of typhoid risk in the population immediately surrounding identified cases. Further studies are needed to understand spatial and temporal trends and to evaluate the effectiveness of targeted vaccination in disrupting typhoid transmission.

### Author summary

New generation typhoid conjugate vaccines have been introduced in mass national campaigns and routine immunization programs in several typhoid-endemic countries. We evaluated evidence for spatial and temporal clustering of typhoid fever in an urban slum area of Dhaka, Bangladesh, where typhoid is highly endemic. We assessed 141 typhoid index cases and observed the highest incidence of typhoid in the 14-day period following case selection in the population residing closest (cluster of radii 0-50-meter) to the index case. Clustering of incident cases suggests a higher intensity of transmission, and the populations at highest risk of contracting typhoid are those living in close proximity to an index case. The risk exists for a period of at least 28 days. Where strong surveillance exists, vaccination of localized geographic areas surrounding cases might be effective in controlling disease outbreaks in low- or no-coverage settings. This approach could offer a practical and efficient strategy for countries with limited data and resources to support mass typhoid vaccination. Additional evaluations of spatial and temporal trends of typhoid transmission are required in endemic and non-endemic settings, and studies designed to evaluate the effectiveness and impact of targeted vaccination are needed to refine prevention and control strategies.

## Introduction

*Salmonella enterica* serotype Typhi (*Salmonella* Typhi) causes potentially severe and occasionally life-threatening disease, referred to as typhoid fever. In 2019, it was estimated that there were 9 million cases of typhoid fever that occurred resulting in 110,000 deaths, primarily in low- and middle-income countries in Asia and Africa [1]. Humans are the only reservoir of *Salmonella* Typhi, and the disease is transmitted through contaminated food and water. Transmission is enabled by inadequate clean water and sanitation facilities and poor personal hygienic practices [2,3], and major outbreaks of typhoid fever have been linked to contaminated municipal water sources [4–7]. Rapid urbanization has resulted in highly populated informal slum settlements within major cities. These slums lack basic infrastructure for

supplying clean water and are therefore ideal grounds for transmission of typhoid and other infectious diseases. Typhoid risk has also been shown to be associated with environmental factors, including proximity to open sewers and highly contaminated water bodies, residence in low elevation areas, and seasons with heavy rainfall [4,8–10]. The distribution of additional risk factors may predict where and when typhoid will occur.

Studies in Iran, Bangladesh, and Uganda have reported high degrees of spatial and temporal clustering of typhoid cases [4,11]. However, limited detailed information describing daily movements, personal contacts, and exposure to other risk factors, confines analysis and understanding of disease transmission dynamics at a community level. Data on the spatial and temporal occurrence of typhoid following an incident case are necessary to design strategies for effective reactive vaccination efforts in vulnerable settings.

In this study, we examined data from a population-based surveillance of enteric fever conducted in a typhoid endemic area in Dhaka, Bangladesh. We investigated whether the risk of incident typhoid following the identification of an index case varied in terms of time and geographic distance among the neighbors of typhoid-infected persons.

## Methods

### Ethics statement

Informed written consent was obtained for all study participants, or from a parent or guardian for minors. Informed assent was obtained for individuals aged 11–17 years. The protocol was approved by the research review committee and ethical review committee of the International Centre for Diarrhoeal Disease Research, Bangladesh (icddr,b) and have been published previously [12].

### Study population

This analysis assessed the population of two wards (Ward 3 and Ward 5) in Mirpur, an urban slum area of Dhaka City, Bangladesh. The original research activities in Bangladesh were conducted as part of a multi-country study (STRATAA), which included Malawi, Nepal, and Bangladesh. Full details of the study have been published elsewhere [13]. In brief, 143,361 residents in the study area were monitored under passive surveillance from August 2016 to April 2018. A census of all individuals residing in the catchment area was conducted at baseline, and demographic information from each household, including GIS (global information system) data points, was collected. Census updates were conducted at 6-month intervals during the study period. We evaluated a dynamic population, accounting for births, in-migrations, out-migrations, and deaths. Members of the study population developing fever were encouraged to seek care at one of several designated study health-care centers. These included five primary level facilities and two inpatient facilities. Patients with a history of fever lasting at least 72 hours or with a measured temperature of $\geq 38.0°C$ at presentation were eligible for inclusion in the study [13]. Blood samples (3mL from patients aged 1-<16 years and 5mL from patients aged $\geq 16$ years) were collected from enrolled participants and were examined by microbiologic culture using an automated system (BD BACTEC Blood Culture System [Becton-Dickinson, Franklin Lakes, NJ, USA] or BacT/ALERT [BioMerieux, Marcy-l'Étoile, France]). For positive cultures, *Salmonella* Typhi organisms were confirmed using colony morphology, Gram staining, standard biochemical tests, and specific antisera. A typhoid case was defined as a participant from within the demographic catchment area, enrolled in the census, and diagnosed with a blood-culture confirmed *Salmonella* Typhi infection.

### Selection of index cases and index controls

All blood-culture confirmed typhoid cases enrolled in the passive surveillance between 15 August 2016 and 15 April 2018 were considered for inclusion in the analysis. Index cases were those that occurred within 28 days before study closure. The selection date of the index case was defined as the date of onset of fever. We selected two, age-group matched (0–4 years, 5–15 years, and 16 years or older) community controls, termed index controls, for each index case at the time of selection. Index controls were randomly selected from census records, ensuring that they were residents of the study area for at least 28 days before the date of selection and had not tested positive for typhoid within 14 days prior to the date of selection. Additionally, controls were selected such that circular regions of radius 200-meter surrounding the index case and both matched index controls could not overlap. The population residing within the circular cluster surrounding index cases and index controls at the time of selection was followed as a cohort for 28 days after selection. Index controls and surrounding clusters populations were not excluded from subsequent selection as cases or controls and index cases could be resampled as index controls after 14 days of case identification.

### Defining spatial clusters and temporal windows

Spatial clusters were defined as mutually exclusive, geometric rings surrounding index cases and index controls with radii scaled from 0 to 200 meters, stepped by 50 meters (0-50-meter, 51-100-meter, 101-150-meter, and 151-200-meter) (Fig 1). We have restricted our follow-up time to 28 days. Temporal windows were defined as the period of 0–14 days and 15–28 days from the date of selection of index cases and matched index controls. The temporal windows were defined on the rough assumption that a single transmission event occurred within 14 days, assuming an average incubation period of typhoid fever of approximately 14 days.

### Evaluation of spatial and temporal clustering

In the evaluation of spatial clustering, we measured the incidence rate (IR) of typhoid fever during the 28 days following index case selection in residents of spatial clusters of increasing distance from the index case and compared these to the IR of residents of similar spatial clusters surrounding index controls. To evaluate temporal clustering, we measured the IR of typhoid fever in cluster residents during the first 14 days after selection (Day 0–14) and the second 14 days after selection (Day 15–28) and compared these rates between index case and index control clusters. We then evaluated both spatial and temporal dimensions simultaneously by comparing the IR in each of the index case spatial clusters versus index control spatial clusters stratified by temporal window (Day 0–14 and Day 15–28).

### Statistical analysis of spatiotemporal risk of typhoid

Cluster residents were evaluated as dynamic populations, accounting for births, in-migrations, out-migrations, and deaths starting at the date of cluster selection. Clusters were observed for 28 days following selection. Person-days (PDs) were calculated for cluster residents from the date of cluster selection to the date of the censoring event, including typhoid fever, migration out of the cluster, or death. IR was calculated as the total number of typhoid cases observed divided by PDs of observation within the cluster. Typhoid cases and PDs of observation could be counted multiple times–as part of an index case cluster and as an incident case within another index case or index control cluster. We applied a Poisson regression model with a log-link function. Person time of the follow-up was considered as an offset and robust standard errors method was used to adjust the p-values and confidence intervals for the design effect

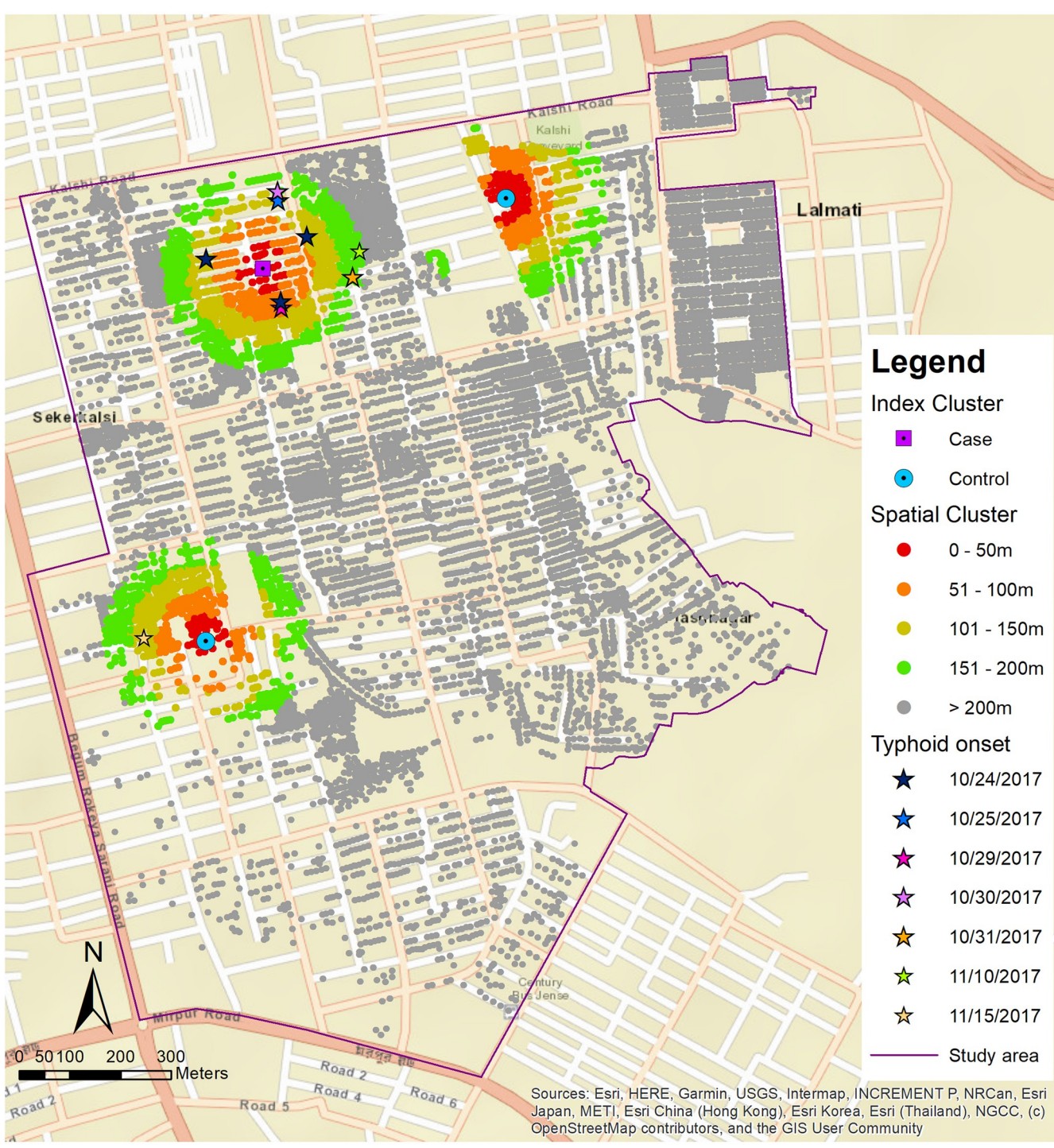

Base map source: Esri, HERE, Garmin, USGS, Intermap, INCREMENT P, NRCan, Esri Japan, METI, Esri China (Hong Kong), Esri Korea, Esri (Thailand), NGCC, (c) OpenStreetMap contributors, and the GIS User Community, https://www.arcgis.com/apps/mapviewer/index.html?layers=3b93337983e9436f8db950e38a8629af
Shapefiles are created and updated by the STRATAA Bangladesh GIS team.

**Fig 1. Geographic representation of the study area and household density.** A typhoid index case and two controls are illustrated and the population residing within concentric ring clusters of increasing diameters surrounding the index case and controls are represented. Incident typhoid cases occurring in the 28-day period following case and control selection are shown. Base map source: Esri, HERE, Garmin, USGS, Intermap, INCREMENT P, NRCan, Esri Japan, METI, Esri China (Hong Kong), Esri Korea, Esri (Thailand), NGCC, (c) OpenStreetMap contributors, and the GIS User Community. Shapefiles are created and updated by the STRATAA Bangladesh GIS team. https://www.arcgis.com/apps/mapviewer/index.html?layers= 3b93337983e9436f8db950e38a8629af.

caused by spatial clusters. Additional adjustments in the models included age in years. The threshold of significance for individual estimates was p<0.05. The sample size was based on 141 index cases and 242 controls selected from the study population. The average population residing within a 50-meter radius circle surrounding index cases and controls was 575. The incidence of typhoid in a 28-day period following selection was 32.5 and 6.6 per 100,000 persons, in case and control clusters respectively. Assuming a coefficient of variation of cluster sizes of 0.50, our analysis had 98.8% power to detect a minimum difference in observed incidence between case and control clusters with 95% confidence. The analysis was performed using the analytical software R (Version 4.2.1), employing the clusterSEs, and sandwich packages to estimate the cluster adjustments confidence interval and p-values [14–16]. ArcGIS 10.8.1 (ESRI Inc.) software was employed to map the spatiotemporal relationship of typhoid within the index clusters. The resulting map was projected in WGS 1984 UTM zone 46N.

## Results

A total of 172 cases of typhoid fever were identified during the study period (Fig 2). Of the total cases, 141 were included as index cases in the analysis. Thirty-one cases occurred within 28 days of the study and were not considered index cases. There were 355 cases of over 80,098,678 PDs within a 200-meter radius during the 28 days follow-up considering redundant counting. The cumulative incidence rate of typhoid fever in the entire population was calculated to be 0.44 per 100,000 PDs (95% CI: 0.40, 0.49). There were no substantive differences in age and sex between cluster populations of cases and controls, however, population density was substantially higher in case clusters (7.6 persons per 10 m$^2$) compared to the control

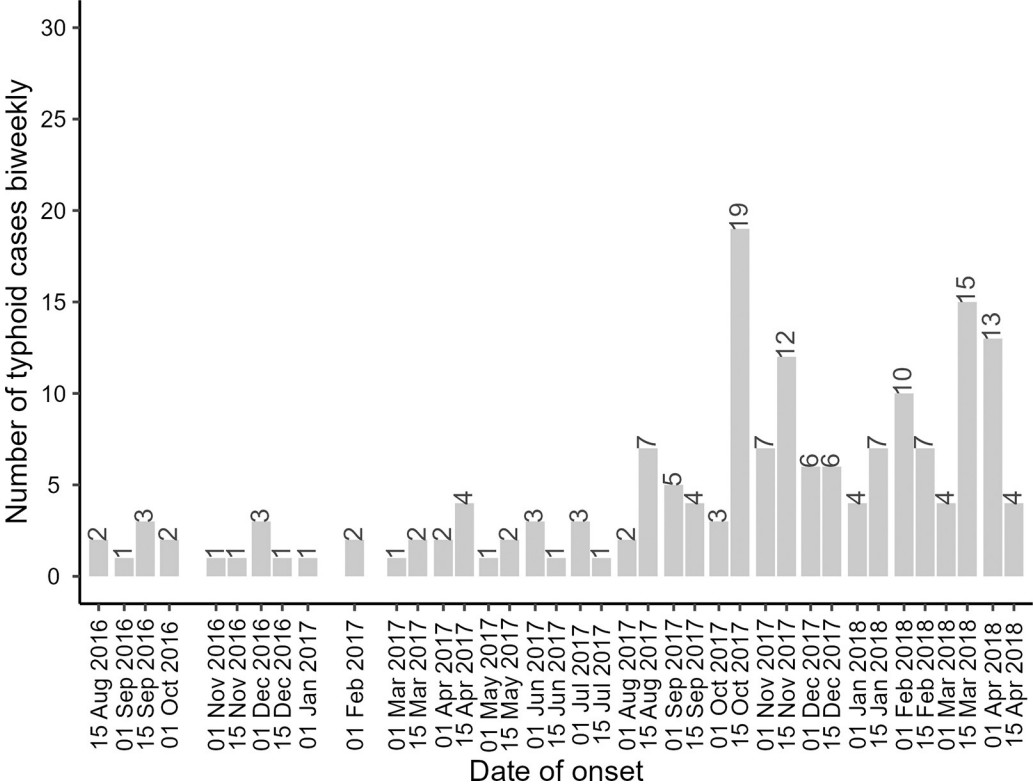

**Fig 2. Number of typhoid cases as reported by study weeks from 15 August 2016 to 15 April 2019, within the study catchment area in Mirpur, Dhaka.**

**Table 1. Baseline characteristics of case and control spatial clusters with a radius of 200-meter.**

| | Index case clusters | Index control clusters |
|---|---|---|
| Males; n (%) | 682751 (50%) | 774125 (50%) |
| Age (in years); mean (SD*) | 25.9 (17.5) | 25.8 (17.5) |
| Cluster population size; median (Q1‡, Q3‡) | 9543 (7745, 12015) | 5123 (2409, 7269) |
| Cluster population density per 10 m²; median (Q1‡, Q3‡) | 7.6 (6.2, 9.6) | 4.1 (1.9, 5.8) |
| Number of households per cluster; median (Q1‡, Q3‡) | 2278 (1873, 2770) | 1198 (572.75, 1691) |

*SD–standard deviation

‡Q1 –first quartile

‡Q3 –third quartile

clusters (4.1 persons per 10 m²) (Table 1). There was evidence of clustering of typhoid cases among the index clusters within 200-meter radius (intra-cluster correlation coefficient: 0.0004; 95% CI: 0.00005–0.0003). Fig 1 illustrates the spatial and temporal distribution of incident cases following an index case identified on 21 October 2017. Two matched index controls were randomly selected at the time of the case selection. Typhoid fever was monitored in the population residing within a 200-meter radius surrounding the index case and controls for a period of 28 days. Concentric rings of increasing radius surrounding the index case and controls are represented by different colors. Eight incident typhoid cases occurred in the index case clusters and one typhoid case occurred in the index control cluster.

## Incidence rates of case and control clusters

During the 28-day follow-up period starting from the index case and control selection date, we counted 268 cases of typhoid within a 200-meter radius of the index case cluster, and 87 typhoid cases in a similar index control cluster (Table 2). The corresponding incidence rates were 0.7 and 0.2 cases per 100,000 PDs of observation in the case and control clusters, respectively. The overall IRR in case clusters compared with control clusters was 3.5 (95% CI: 2.7, 4.4).

## Comparison of typhoid incidence rates in spatial clusters

In the innermost spatial cluster with a 50m radius surrounding the index case and controls, the typhoid incidence over a 28-day period was 1.2 per 100,000 PDs (95% CI: 0.8, 1.6) for residents in case clusters and 0.2 per 100,000 PDs (95% CI: 0.1, 0.5) for residents in control clusters. The corresponding adjusted IRR was 4.9 (95% CI: 2.4, 10.3) (Table 2). In the outermost spatial cluster (151-200-meter), we observed a typhoid incidence of 0.7 per 100,000 PDs (95% CI: 0.5, 0.8) for case clusters and 0.2 per 100,000 PDs (95% CI: 0.1, 0.3) for control clusters. The adjusted IRR in the outermost cluster was 3.2 (95% CI: 2.2, 4.8) (Table 2).

Relative to typhoid incidence in residents of the innermost index case clusters with radius 50-meter, the IRR in residents of progressively larger spatial index case clusters were 0.7 (95% CI: 0.4, 1.0), 0.6 (95% CI: 0.4, 0.8), and 0.6 (95% CI: 0.4, 0.9), in clusters 51-100-meter, 101-150-meter, and 151-200-meter respectively (Table 3).

## Comparison of typhoid incidence rates in spatial clusters stratified by time

We compared the incidence rate of typhoid in residents of index case and control clusters during the first 14 days after selection and the second 14 days after selection. During days 0–14, the IRR of typhoid between residents of index case clusters and control clusters ranged

**Table 2. Comparison of typhoid incidence during 28 days of follow-up by varying spatial size of index case and index control clusters.**

| Spatial clusters | Population; person-days | Typhoid cases | IR* per 100,000 person-days, (95% CI†) | Crude IRR‡(95% CI) | Adjusted§ IRR (95% CI) |
|---|---|---|---|---|---|
| **0-200-meter** | | | | | |
| Index control cluster | 1538967; 42520780 | 87 | 0.2 (0.2, 0.25) | Ref. | Ref. |
| Index case cluster | 1357330; 37577898 | 268 | 0.7 (0.6, 0.8) | 3.5 (2.7, 4.4) | 3.5 (2.7, 4.4) |
| **0-50-meter** | | | | | |
| Index control cluster | 137790; 3798884 | 9 | 0.2 (0.1, 0.5) | Ref. | Ref. |
| Index case cluster | 108762; 3011137 | 35 | 1.2 (0.8, 1.6) | 4.9 (2.4, 10.2) | 4.9 (2.4, 10.3) |
| **51-100-meter** | | | | | |
| Index control cluster | 341941; 9442930 | 21 | 0.2 (0.1,0.3) | Ref. | Ref. |
| Index case cluster | 275893; 7631715 | 57 | 0.8 (0.6, 1.0) | 3.4 (2.0, 5.5) | 3.4 (2.0, 5.6) |
| **101-150-meter** | | | | | |
| Index control cluster | 477424; 13202015 | 24 | 0.18 (0.1, 0.3) | Ref. | Ref. |
| Index case cluster | 427544; 11838797 | 76 | 0.6 (0.5, 0.8) | 3.5 (2.2, 5.6) | 3.5 (2.2,5.5) |
| **151-200-meter** | | | | | |
| Index control cluster | 581812; 16076951 | 33 | 0.2 (0.1, 0.3) | Ref. | Ref. |
| Index case cluster | 545131; 15096249 | 100 | 0.7 (0.5, 0.8) | 3.2 (2.2, 4.8) | 3.2 (2.2, 4.8) |

*IR–incidence rate

‡CI-confidence interval

‡IRR–incidence rate ratio

§Adjusted by age in years and clustering by spatial clusters

between 4.9 (95% CI: 1.8, 13.0) in the innermost clusters with radius 0-50m and 3.0 (95% CI: 1.7, 5.1) in the outermost clusters with radius 151-200m. During days 15–28, the IRR of typhoid ranged between 5.1 (95% CI: 1.7, 15.2) in the innermost clusters and 3.6 (95% CI: 2.0, 6.3) in the outermost cluster (Table 4).

**Table 3. Relative rate ratio of typhoid incidence in progressively distant spatial case index clusters compared to the closest spatial case index cluster.**

| Index case clusters | Population /person-days | Typhoid cases | IR* per 100,000 person-days (95% CI†) | Crude IRR‡ (95% CI) | Adjusted IRR§ (95% CI) |
|---|---|---|---|---|---|
| **0-50-meter** | 108762 /3011137 | 35 | 1.2 (0.8, 1.6) | Ref. | Ref. |
| **51-100-meter** | 275893 /7631715 | 57 | 0.8 (0.6, 1.0) | 0.6 (0.4, 1.0) | 0.7 (0.4, 1.0) |
| **101-150-meter** | 427544 /11838797 | 76 | 0.6 (0.5, 0.8) | 0.6 (0.4, 0.8) | 0.6 (0.4, 0.8) |
| **151-200-meter** | 545131 /15096249 | 100 | 0.7 (0.5, 0.8) | 0.6 (0.4, 0.8) | 0.6 (0.4, 0.9) |

*IR–incidence rate

†CI-confidence interval

‡IRR–incidence rate ratio

§Adjusted by age in years

**Table 4. Comparison of typhoid incidence by temporal window between index case and index control clusters.**

| | 0–14 days after selection | | | | 15–28 days after selection | | | |
|---|---|---|---|---|---|---|---|---|
| | Population; person-days | Cases | IR* per 100,000 person-days | Adjusted§ IRR‡ (95% CI†) | Population; person-days | Cases | IR per 100,000 person-days | Adjusted§ IRR‡ (95% CII†) |
| **0-50-meter** | | | | | | | | |
| Index control cluster | 137790; 1912892 | 5 | 0.3 | Ref. | 135566; 1886062 | 4 | 0.2 | Ref. |
| Index case cluster | 108762; 1514277 | 19 | 1.3 | 4.9 (1.8, 13.0) | 107358; 1497085 | 16 | 1.1 | 5.1 (1.7, 15.2) |
| **51-100-meter** | | | | | | | | |
| Index control cluster | 341941; 4755079 | 10 | 0.2 | Ref. | 337006; 4687977 | 11 | 0.2 | Ref. |
| Index case cluster | 275893; 3837986 | 26 | 0.7 | 3.2 (1.6, 6.7) | 272241; 3794065 | 31 | 0.8 | 3.5 (1.8, 7.0) |
| **101-150-meter** | | | | | | | | |
| Index control cluster | 477424; 6644208 | 15 | 0.2 | Ref. | 471310; 6557989 | 9 | 0.1 | Ref. |
| Index case cluster | 427544; 5953057 | 41 | 0.7 | 3.0 (1.7, 5.5) | 422566; 5886258 | 35 | 0.6 | 4.3 (2.1, 8.9) |
| **151-200-meter** | | | | | | | | |
| Index control cluster | 581812; 8093132 | 18 | 0.2 | Ref. | 573855; 7984015 | 15 | 0.2 | Ref. |
| Index case cluster | 545131; 7590417 | 50 | 0.7 | 3.0 (1.7, 5.1) | 538972; 7506462 | 50 | 0.7 | 3.6 (2.0, 6.3) |

*IR–incidence rate

†CI-confidence interval

‡IRR–incidence rate ratio

§Adjusted by age in years and clustering by spatial clusters

## Discussion

In a typhoid endemic, urban setting, we evaluated evidence for spatial and temporal clustering of typhoid fever. Over a period of 20 months, we assessed 141 typhoid index cases and observed the highest incidence of typhoid in the 28-day period following case selection in the population residing closest (cluster of radii 0-50-meter) to the index case. The clustering of incident cases is suggestive of higher intensity of transmission, although our analysis was not designed to distinguish direct transmission events. Typhoid incidence decreased in populations residing in progressively increasing geographic distances from the index case, although the incidence was sustained even in the outermost clusters with a radius of 150-200-meter. The difference in typhoid incidence in the first 14 days versus the second 14 days following an index case was negligible between case and control clusters at all spatial levels evaluated. These findings suggest that the populations at highest risk of contracting typhoid are those living in close proximity to an index case and that the risk remains constant up to a period of at least 28 days.

Spatial and temporal clustering of typhoid have been reported in relation to ecological factors. Environmental and socio-economic factors determine the spatial distribution of typhoid, concentrating in areas with poor sanitation and high population density [8]. Temporal clustering aligns with seasonal variations and is impacted by temperature and rainfall [17]. Several studies have reported on the spatiotemporal trends of typhoid fever in different settings [11,18–21] and phylogenetic analyses of *Salmonella* Typhi isolates have aided in the understanding of modes of transmission in endemic areas [18]. However, these approaches did not

empirically evaluate the comparative risk of typhoid at varying degrees of spatial and temporal distance from an index case.

Safe and highly efficacious new generation conjugate typhoid vaccines are now available and have been introduced in mass national campaigns and routine immunization programs in several countries [22]. However, lack of national-level, systematic, blood-culture confirmed typhoid burden data in many endemic countries slows vaccine introduction efforts, despite availability and support through Gavi programs [22]. As an interim control strategy, our findings lend support for a localized approach, such as case area targeted index (CATI) vaccination, where individuals residing in geographic areas surrounding cases are vaccinated reactively following case detection. There are, however, limitations to this approach. Humoral and cellular-mediated immune responses sufficient to protect against *Salmonella* Typhi infection takes several weeks to develop, and therefore CATI vaccination campaigns may fail to prevent onward transmission from case in the incubatory phase before the campaign starts. There are several important limitations to consider when interpreting these results. Firstly, spatial clusters were delineated based on arbitrary geometric cutoffs, which may not represent practical epidemiologic boundaries of disease transmission. Our analysis assumed that for incident cases, the household represented the location where typhoid was transmitted or acquired. However, transmission or introduction of typhoid from outside defined spatial clusters would make our analysis conservative. Secondly, typhoid diagnostics are insensitive, and incident cases in index case and control clusters may be associated with transmission from unidentified or asymptomatic cases. Additional analyses, including the phylogenetic comparison of typhoid isolates, were not conducted to better understand transmission pathways, therefore it was not possible to determine if incident typhoid cases were linked to the index case. Thirdly, our analytic approach permitted index cases to be assessed as incident cases for neighboring index cases, resulting in case redundancy. However, this redundancy does not affect our estimated incidence rate and incidence rate ratio due to the redundancy of person-time in both index case and control clusters. Fourthly, we evaluated a 28-day period following selection. This period may not have been long enough to observe important incident patterns associated with a typhoid case that occurs at a longer time scale. The incubation period of typhoid fever can range from 6 to 30 days [23,24], therefore transmission events may occur beyond the window of observation. However, for public health intervention, we were interested in evaluating a period that might reflect the time frame within which a reactive targeted campaign might be effectively implemented. Finally, the analysis did not consider other epidemiological variables, such as population density, and the presence of a common piped water grid, which could potentially confound the incidence of the disease. We observed a significant difference in population density between cases and controls. The geographic requirement that clusters surrounding cases and controls do not overlap may have introduced bias based on the occurrence of typhoid in high-dense regions and the exclusion of these regions from control selection. Despite these limitations, our findings are strengthened by the reliance on a well-established clinical surveillance team and a highly experienced technical laboratory. The population under observation was highly sensitized to seek care for febrile illness at a study center, therefore the impact of variance in healthcare seeking behavior and missed cases is minimal.

Our study uniquely evaluates the spatial and temporal patterns of incident cases following an index case in a typhoid endemic setting. Where strong surveillance exists, vaccination of localized geographic areas surrounding cases might be effective in limiting disease outbreaks in low-coverage or no coverage settings. This approach, while imperfect, offers a practical strategy to combat disease in the absence of a national TCV immunization strategy. Additional evaluations of spatial and temporal trends of typhoid transmission are required in endemic

and non-endemic settings and studies designed to evaluate the effectiveness and impact of targeted vaccination are needed to refine prevention and control programs.

## Acknowledgments

We acknowledge all the investigators and contribution of the STRATAA study consortium and the participants who had taken part in the large field and laboratory teams at the sites.

## Author Contributions

**Conceptualization:** Faisal Ahmmed, Farhana Khanam, Md Taufiqul Islam, Firdausi Qadri, John D. Clemens, Birkneh Tilahun Tadesse, Justin Im.

**Data curation:** Faisal Ahmmed, Birkneh Tilahun Tadesse.

**Formal analysis:** Faisal Ahmmed, Deok Ryun Kim, Md Golam Firoj, Xinxue Liu, Justin Im.

**Funding acquisition:** Firdausi Qadri, John D. Clemens.

**Investigation:** Faisal Ahmmed, Farhana Khanam, John D. Clemens, Birkneh Tilahun Tadesse, Justin Im.

**Methodology:** Faisal Ahmmed, Farhana Khanam, John D. Clemens, Justin Im.

**Project administration:** Farhana Khanam, Firdausi Qadri, John D. Clemens.

**Resources:** Farhana Khanam, Firdausi Qadri, John D. Clemens.

**Software:** Faisal Ahmmed.

**Supervision:** Khalequ Zaman, Florian Marks, Jerome H. Kim, Firdausi Qadri, John D. Clemens.

**Validation:** Faisal Ahmmed, Farhana Khanam, Md Taufiqul Islam, John D. Clemens, Birkneh Tilahun Tadesse, Justin Im.

**Visualization:** Faisal Ahmmed, Farhana Khanam, Md Taufiqul Islam, Deok Ryun Kim, Sophie Kang, Suman Kanungo, Birkneh Tilahun Tadesse, Justin Im.

**Writing – original draft:** Faisal Ahmmed, Farhana Khanam, Md Taufiqul Islam, Birkneh Tilahun Tadesse, Justin Im.

**Writing – review & editing:** Faisal Ahmmed, Farhana Khanam, Md Taufiqul Islam, Asma Binte Aziz, Masuma Hoque, Xinxue Liu, Hyon Jin Jeon, Suman Kanungo, Fahima Chowdhury, Ashraful Islam Khan, Florian Marks, Jerome H. Kim, Firdausi Qadri, John D. Clemens, Birkneh Tilahun Tadesse, Justin Im.

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
