## [Decision Letter · Decision Letter 0]

22 Dec 2023

Dear Mr. Ahmmed,

Thank you very much for submitting your manuscript "Spatial and Temporal Clustering of Typhoid Fever in an Urban Slum of Dhaka City: Implications for Targeted Typhoid Vaccination" for consideration at PLOS Neglected Tropical Diseases. As with all papers reviewed by the journal, your manuscript was reviewed by members of the editorial board and by several independent reviewers. In light of the reviews (below this email), we would like to invite the resubmission of a significantly-revised version that takes into account the reviewers' comments. 

We cannot make any decision about publication until we have seen the revised manuscript and your response to the reviewers' comments. Your revised manuscript is also likely to be sent to reviewers for further evaluation.

Sincerely,

Epco Hasker

Academic Editor

Stuart Blacksell

Section Editor

Reviewer's Responses to Questions

**Key Review Criteria Required for Acceptance?**

**Methods**

-Are the objectives of the study clearly articulated with a clear testable hypothesis stated?

-Is the study design appropriate to address the stated objectives?

-Is the population clearly described and appropriate for the hypothesis being tested?

-Is the sample size sufficient to ensure adequate power to address the hypothesis being tested?

-Were correct statistical analysis used to support conclusions?

-Are there concerns about ethical or regulatory requirements being met?

Reviewer #1: The authors have clearly reported the methodology including the study setting, selection of cases and matched controls and marking the geographical areas around index cases and controls. Sample size estimation is not reported.

Reviewer #2: The authors aimed to provide further understanding of transmission of typhoid fever linked to index case identified in a high risk setting, a slum in this study and over a time frame that the authors defined. The objectives are clear and the population under study has been described sufficiently to answer the question under investigation. The authors investigated clusters of typhoid transmission linked to an index case and also to index controls.

Reviewer #3: • Ethical approvals and consent processes are in reference 12, but typically they should be explicitly stated in the manuscript.

• How were controls approached, recruited, and enrolled? Please briefly elaborate on the process.

• Per PNTD checklist, please explicitly state the study objectives and include a clearly stated hypothesis.

• Since the STRATAA study (ref 12) serves as the backbone of the study population, a brief summary of that study would be helpful in Background for appropriate context.

• Study population: The study population described in Lines 77-87 appears different from that in ref 12. This study cites ~158,000 residents versus ~110,000 residents in ref 12 Figure 1. This study surveys from August 2016 through April 2018 (I believe it is from Sept 2016 based on Fig 2) versus passive surveillance dates in ref 12 from Jan 1, 2017 – Dec 31 2018. This study identified 172 total cases during the 20-month study period (Line 130) versus 359 S. Typhi cases in Dhaka over 24 different months in ref 12 Figure 1. Please either describe the study population, or explain any differences compared with ref 12, especially with regard to census population and study dates as these numbers greatly impact the case counts and statistical analyses on incidence and incidence ratios. 

• Lines 84-86: Please briefly elaborate on the sample acquisition protocols and “microbiological culture methods.” I see reference #12, but some basic details should be present in the Methods section. For example, were two sets of bloods routinely drawn? Was blood culture performed by an automated blood culture machine – if so, what machine? How was S. Typhi isolated and identified (broths, agars, slants, biochemical strips, agglutination, PCR, sequencing, etc.)?

• Line 89: Were the same typhoid cases that were used as index cases then counted in relation to another index case? Were any cases counted more than once?

• Line 105: Please offer a brief biological (or other) rationale for selecting time windows of 0-14 vs 15-28 days post infection. What do these windows represent or what epidemiological questions can they be expected to answer?

• The manuscript would greatly benefit from a map (or maps by Ward) of the geographic region with points to indicate the index and control households and surrounding ring of analysis. Added layers for waterways, water piping, relevant sewerage, and/or topographical boundaries could be especially useful as well. These details are essential for a reader to visualize the spatial and temporal clustering analysis. Regarding human subject confidentiality, typically if large markers are used, if the map is zoomed far enough out, and if there is a high background density of people (5,000-10,000 people within any given 200m radius), the identity of individuals can remain adequately preserved.

• Why did the authors select the method of choice? Did you consider a simple point-pattern cluster or density analysis of all households over the study period to further investigate clustering? Or a space-time scan to evaluate temporal-spatial relationships? Arcgis and satscan softwares have a lot of powerful options.

Reviewer #4: Methods are generally clear and straightforward. Some more direct formal comparisons of statistical difference/power would be nice.

**Results**

-Does the analysis presented match the analysis plan?

-Are the results clearly and completely presented?

-Are the figures (Tables, Images) of sufficient quality for clarity?

Reviewer #1: All-important results are clearly narrated. Tables and figures are adequate.

Reviewer #2: The authors have clearly presented the results. The figures and the Tables are clear and of good quality. The results clearly indicate increased transmission in areas in close proximity to the index case and therefore case area targeted interventions would be useful. 

I have a few questions and also comments for the authors to consider in the discussion

1)

In line 100 under 'Selection of cases and controls' the authors indicate that cases could also be resampled as controls. How many of the cases were resampled as controls if any, and how does this resampling affect the assertion that the population density was higher in cases than it was in controls?

2)

Figure 2 shows the number of typhoid cases observed bi-weekly during study period. There is an increase in the number of cases identified from August 2017. Is this increase reflected in both controls and case clusters? What factor may have resulted in the increased observations and also how does this factor affect spatial clusters?

3) Under limitations the authors have indicated the 28 day duration as not sufficient to assess transmission dynamics. Perhaps the 200M radius may also not be optimal to observe a changes in transmission. According to TABLE 2 typhoid cases were identified along the radius regardless of spatial size in both cases and controls.

Reviewer #3: • Line 131: Were any of the 31 cases occurring within 28 days before study end included as case counts if they fell 200m from an index case occurring 28 days before study end? I ask because 31 cases in one month is a sizable burden of disease when 172 total cases over 20 months averages to ~8.6 cases per month. Did an outbreak occur? Were these 31 cases close to one another or on a common piped water grid?

• Line 132: How was the incidence of 0.44 cases per 100,000 person-days calculated? By my calculation, if 141 cases occurred over 20 months in a population of 158,000 people, the incidence per 100,000 person-days should be:

141 cases / (158000 persons * (20 months * 365 days / 12 months) ) * 100000 = 0.15 cases per 100,000 person-days

 Please share how you calculated incidence. 

• Lines 134-135: Cluster population size and density were substantially higher in case clusters (Table 1). Given the mechanisms of typhoid transmission, would this difference confound your results? Please address this in Discussion.

• Line 139: 268 observed cases is above the 141 cases reported during the study period. Were cases counted multiple times? If so this should be explained in methods. A map of points and geometric rings as requested above would also assist in visualizing this. Does this impact your analysis?

• Figure 2: X-axis suggests the study is “from Sept 2016 through Apr 2018.” Versus “from August 2016” as is stated in in Line 79.

• I do not see a table that matches Line 112, “To evaluate temporal clustering, we measured the IR of typhoid fever in cluster residents (r=200m) during the first 14 days after selection […]” Table 4 is the closest, but makes comparisons at 50m increments, rather than r=200m. Was there a comparison of all cases from 0-200m at 0-14d vs 15-28d?

Reviewer #4: Results are not clearly presented.

**Conclusions**

-Are the conclusions supported by the data presented?

-Are the limitations of analysis clearly described?

-Do the authors discuss how these data can be helpful to advance our understanding of the topic under study?

-Is public health relevance addressed?

Reviewer #1: The study conclusion is in line with the study objective and main findings.

Reviewer #2: Conclusions are supported by the data presented and the public health relevance is clear as regards case area targeted interventions including vaccines as authors recommend but also other interventions should be considered.

Reviewer #3: • The discussion section needs improvement. A scholarly discussion with references on the meaning and significance of the results is necessary. Contextualize the study within the local typhoid dynamics (and literature). Relate the findings to other studies on the spatial and temporal nature of typhoid. 

• Lines 180-188: Other limitations include use of household coordinates (which may not equate to the place where typhoid was transmitted/acquired), counting of cases multiple times (if they fall into multiple 200m radii), and perhaps a lack of other epidemiologic data (including household level data of cases, familial relationships, water/sewage data, knowledge of carriers in the area, etc.)

• Line 188: Change “transmission events,” to incident patterns or similar. The study is not presently designed to evaluate transmission patterns.

• Lines 190-194: These three sentences seem to repeat the data summarized in lines 171-176. If they do not add information, they can probably be removed to avoid redundancy.

• Lines 195-196: Similar to the comment above for Line 188, I don’t think the data supports a transmission observation so much as the existence of co-incident cases occurring near one another.

• Lines 196-200: “Reactive vaccination” and “case-area targeted intervention (CATI) strategies” are introduced for the first times in the penultimate sentence of the manuscript. If a discussion point, please define and elaborate your idea. Why do you think the data supports a “reactive vaccination” campaign as the best intervention strategy? Why not mass vaccination campaigns? Or routine/EPI immunizations? More discussion is warranted with appropriate references.

Reviewer #4: Conclusions are not well supported by data presented-- more is needed in discussion.

**Editorial and Data Presentation Modifications?**

Reviewer #1: The authors are advised to report the sample size estimation and also to further enhance the discussion section.

Reviewer #2: (No Response)

Reviewer #3: • Line 57: comma after “S. Typhi”

• Line 62: disease should be plural

• Line 67: the word “and” may be erroneous or some words are missing

• Line 83: Are the 7 participating healthcare centers inclusive of ALL the healthcare centers with blood culture capability serving the ~158,000 residents? If not, then the population surveilled may be less.

• Line 87: “typhi” should be capitalized

• Lines 99-101: Were any index controls selected more than once?

• Line 142: missing a space

• Line 152: missing a space

• Line 157: missing an open parenthesis

• Lines 175-176: seems contradictory

• Lines 190-194: Redundant to lines 171-178. Not adding any additional discussion.

• References formatting is irregular. See the way that authors are listed, for example

Reviewer #4: My main concern is that there needs to be more in the discussion. The first paragraph of the discussion reads a little more like how the results should actually be (or, at best, is just a re-hash of those results). In discussion, it would be good to talk more about:

1) What do you think explains these data results—spatial and temporal

2) What hypotheses of transmission are supported or not supported by these findings

3) How these results may be extrapolated to other settings, or not extrapolated, based on what you know of this area versus other typhoid endemic areas. 

4) Are these cases mostly sporadic, clusters, outbreaks? Are any of these household cases? Do you know?

5) More about reactive vaccination campaigns (perhaps needs to be more in background too)—it’s a leap to just show results and then say this in a conclusion without any discussion or further support. How do each of your conclusions (spatial, and temporal) each independently impact recommendations? Does lack of temporal difference mean that reactive vaccination doesn’t have to be immediate? That disease risk is more about specific, highly focal areas with specific environmental risks, or about risk generated following individual cases, or what?

**Summary and General Comments**

Reviewer #1: This is an important study with sufficient rigors and novelty. Data analysis is adequately performed. Sample size estimation is missing which should be included. Further, discussion is inadequate and recomm

---

## [Decision Letter · Decision Letter 1]

4 Apr 2024

Dear Mr. Ahmmed,

Thank you very much for submitting your manuscript "Spatial and Temporal Clustering of Typhoid Fever in an Urban Slum of Dhaka City: Implications for Targeted Typhoid Vaccination" for consideration at PLOS Neglected Tropical Diseases. As with all papers reviewed by the journal, your manuscript was reviewed by members of the editorial board and by several independent reviewers. The reviewers appreciated the attention to an important topic. Based on the reviews, we are likely to accept this manuscript for publication, providing that you modify the manuscript according to the review recommendations.

Sincerely,

Epco Hasker

Academic Editor

Stuart Blacksell

Section Editor

Reviewer's Responses to Questions

**Key Review Criteria Required for Acceptance?**

**Methods**

-Are the objectives of the study clearly articulated with a clear testable hypothesis stated?

-Is the study design appropriate to address the stated objectives?

-Is the population clearly described and appropriate for the hypothesis being tested?

-Is the sample size sufficient to ensure adequate power to address the hypothesis being tested?

-Were correct statistical analysis used to support conclusions?

-Are there concerns about ethical or regulatory requirements being met?

Reviewer #2: The authors have now provided extra clarity to the methods section

Reviewer #3: The authors comprehensively and adequately addressed prior methodologic concerns/questions.

Very nice incorporation of revised Fig 1 to visualize the methods employed.

Reviewer #4: The population (and geography) continues to be incompletely described; A map showing the area, including where cases were and controls and underlying pop density would help. How large even is the study region? Very relevant because we are talking about geospatial analyses here. It is not clear to me if the exclusion criteria for controls are incompletely described, given the stark differences in pop density surrounding controls (which coudl be random, but seem so different as to suggest a potential systematic reason).

**Results**

-Does the analysis presented match the analysis plan?

-Are the results clearly and completely presented?

-Are the figures (Tables, Images) of sufficient quality for clarity?

Reviewer #2: The results are clearly presented

Reviewer #3: Analysis and results are clearly and completely presented in the revised manuscript.

Reviewer #4: In general, the results adequately describe what is proposed. An additional map to clarify study pop and methods would be helpful. There is one map on page 46 of the pdf, but it is not interpretable and has no captions and otherwise appears to provide very limited information. A simple map showing underlying population density, the study area, and where cases and controls (perhaps with 200m circles, if visually doable) were would probably be very helpful.

**Conclusions**

-Are the conclusions supported by the data presented?

-Are the limitations of analysis clearly described?

-Do the authors discuss how these data can be helpful to advance our understanding of the topic under study?

-Is public health relevance addressed?

Reviewer #2: The conclusions discussed are supported by the findings from the study.

Reviewer #3: The authors expanded the discussion greatly and with added nuance. For example, the inclusion of "our analysis was not designed to distinguish direct transmission events" is important. An alternative explanation for the spatial clustering is that a source outbreak (e.g., contaminated food shared, or water reaching a small region) occurred and infected individuals develop typhoid at variable rates, with some occurring <14 days and others up to 30 days.

The authors also greatly improved discussion of study limitations and context.

Two final considerations regarding the recommendation for reactive vaccination:

1) Based on the data presented, the authors recommend CATI; however I am not convinced this is a perfect solution. This control method assumes that a targeted vaccination following identification of an index vaccine would prevent cases occurring presumably ~14-28 days after the index case. However, development of humoral and cellular-mediated immune protection sufficient to prevent infection following challenge with S. Typhi is at least several weeks, and some individuals receiving CATI vaccine may be incubatory cases from the same source of infection, not yet presenting with symptoms. This nuance is sufficiently reflected in the limitations discussed and does not necessarily need to be further elaborated upon.

2) The use of CATI without antecedent case/household investigations could obscure the ability for public health teams to identify chronic S. Typhi carriers using serological methods. For example, one proposed tool to detect chronic carriers (which may be a source and longterm reservoir of transmission in these high incident communities) is anti-Vi IgG antibody, among other serological markers of chronic infection. It might be beneficial to perform household investigations of incident cases and collect blood/stool specimens from potential chronic carriers PRIOR to vaccinating them with typhoid conjugate vaccine, which would induce high titers of anti-Vi IgG antibody in vaccinees, losing the ability to detect chronic carriers possessing uniquely high titers compared to household members. 

I would leave it to the authors to consider the above in their recommendations for CATI approaches, but changes are not requisite as these are areas of ongoing research.

Reviewer #4: No-- I don’t think the conclusion is well supported by or discussed in depth in the paper. “In countries lacking mass national campaigns and routine immunization programs, evidence of geographic clustering of typhoid cases suggests the potential impact of typhoid reactive vaccination in the population immediately surrounding identified cases.” How so? Is Bangladesh one of these countries? Why only countries lacking mass national campaigns? What is done now? What about the time needed to vaccinate and to then develop immunity? How does that relate to the limitations in timelines of this study? I understand what the authors are trying to say, and there is more here now then before, but I think this statement is overly broad, and there continues to not be a lot of discussion around these points. If this is a central conclusion (as it appears to be) there should be more around this is discussion if this is the conclusion. Otherwise, the conclusion should state only what is clearly supported by the evidence—which is not around efficacy of different vaccine strategies.

**Editorial and Data Presentation Modifications?**

Reviewer #2: None

Reviewer #3: Change "sever" to "several" (Methods > Study Population > Line 109 in complete PDF)

Reviewer #4: 66-67: run-on sentence

68: This sentence, “A better understanding of transmission linked to an identified typhoid fever case in a slum setting is needed” does not describe what this paper tries to do. This paper is looking at risk associated with spatiotemporal factors, not transmission factors of individual cases. 

83: what kind of health center—inpatient or outpatient? Pharmacy?

**Summary and General Comments**

Reviewer #2: This is an important study that provides a platform for future studies designed to investigate transmission dynamics

Reviewer #3: Ahmmed et al. effectively present a simple spatial and temporal risk analysis of typhoid fever occurring in two wards of Mirpur, Dhaka City, Bangladesh. The revisions to the manuscript are comprehensive and addressed all my previous comments adequately.

Reviewer #4: Overall, the revisions have been very helpful and this is a much stronger paper. The increased delineation of methods and results have helped make the study much clearer. However, important concerns remain, including around the populations included for control/cases, as well as the conclusions reached that are not really discussed.

Major issues:

A major difference between cases and controls is the density; these are marked differences. Why is it? This does not seem likely to be random – is it because of the exclusion criteria (non overlapping with cases), or some other reason? This should also be addressed in limitations.

How does inclusion of household clusters, in the setting of few overall cases, bias these results? A description of the household clusters (or otherwise directly associated cases) should be included. With only 141 cases over 20 months, it could easily be that one or two households with cases of say 5 each, would explain most of your results by heavily weighting your inner most circle multiple times (household case 1 would have all 4 other people in house count toward innermost incidence rate; household case 2 would then have case 1 and the other 3 people count; household case 3 would have case 1 and 2 and the other 2 people count, etc etc). Was a sensitivity analysis done excluding immediate household members/cases from analysis—this would strengthen your conclusion that a vaccine campaign should focus on close geography (rather than just immediate household members, as is currently often the recommendation in many places).

No-- I don’t think the conclusion is well supported by or discussed in depth in the paper. “In countries lacking mass national campaigns and routine immunization programs, evidence of geographic clustering of typhoid cases suggests the potential impact of typhoid reactive vaccination in the population immediately surrounding identified cases.” How so? Is Bangladesh one of these countries? Why only countries lacking mass national campaigns? What is done now? What about the time needed to vaccinate and to then develop immunity? How does that relate to the limitations in timelines of this study? I understand what the authors are trying to say, and there is more here now then before, but I think this statement is overly broad, and there continues to not be a lot of discussion around these points. If this is a central conclusion (as it appears to be) there should be more around this is discussion if this is the conclusion. Otherwise, the conclusion should state only what is clearly supported by the evidence—which is not around efficacy of different vaccine strategies.

PLOS authors have the option to publish the peer review history of their article (what does this mean?). If published, this will include your full peer review and any attached files.

Reviewer #2: Yes: Chisomo Msefula

Reviewer #3: No

Reviewer #4: No

Figure Files:

Data Requirements:

Reproducibility:

References

---

## [Editor Report · Decision Letter 2]

21 May 2024

Dear Mr. Ahmmed,

Thank you very much for submitting your manuscript "Spatial and Temporal Clustering of Typhoid Fever in an Urban Slum of Dhaka City: Implications for Targeted Typhoid Vaccination" for consideration at PLOS Neglected Tropical Diseases. As with all papers reviewed by the journal, your manuscript was reviewed by members of the editorial board and by several independent reviewers. The reviewers appreciated the attention to an important topic. Based on the reviews, we are likely to accept this manuscript for publication, providing that you modify the manuscript according to the review recommendations. 

Sincerely,

Epco Hasker

Academic Editor

Stuart Blacksell

Section Editor

Figure Files:

Data Requirements:

Reproducibility:

References

---

## [Editor Report · Decision Letter 3]

6 Jun 2024

Dear Ahmmed,

We are pleased to inform you that your manuscript 'Spatial and Temporal Clustering of Typhoid Fever in an Urban Slum of Dhaka City: Implications for Targeted Typhoid Vaccination' has been provisionally accepted for publication in PLOS Neglected Tropical Diseases.

Best regards,

Epco Hasker

Academic Editor

Stuart Blacksell

Section Editor

---

## [Editor Report · Acceptance letter]

20 Jun 2024

Dear Ahmmed,

We are delighted to inform you that your manuscript, "Spatial and Temporal Clustering of Typhoid Fever in an Urban Slum of Dhaka City: Implications for Targeted Typhoid Vaccination," has been formally accepted for publication in PLOS Neglected Tropical Diseases.

Best regards,

Shaden Kamhawi

co-Editor-in-Chief

Paul Brindley

co-Editor-in-Chief
